# Cameroonian Spice Extracts Modulate Molecular Mechanisms Relevant to Cardiometabolic Diseases in SW 872 Human Liposarcoma Cells

**DOI:** 10.3390/nu13124271

**Published:** 2021-11-26

**Authors:** Achille Parfait Atchan Nwakiban, Anna Passarelli, Lorenzo Da Dalt, Chiara Olivieri, Tugba Nur Demirci, Stefano Piazza, Enrico Sangiovanni, Eugénie Carpentier-Maguire, Giulia Martinelli, Shilpa Talkad Shivashankara, Uma Venkateswaran Manjappara, Armelle Deutou Tchamgoue, Gabriel Agbor Agbor, Jules-Roger Kuiate, Maria Daglia, Mario Dell’Agli, Paolo Magni

**Affiliations:** 1Department of Biochemistry, Faculty of Science, University of Dschang, Dschang P.O. Box 96, Cameroon; achilestyle@yahoo.fr (A.P.A.N.); jrkuiate@yahoo.com (J.-R.K.); 2Department of Pharmacological and Biomolecular Sciences, Università degli Studi di Milano, 20133 Milan, Italy; anna.passarelli2393@gmail.com (A.P.); lorenzo.dadalt@unimi.it (L.D.D.); chia.olivieri@gmail.com (C.O.); tugba.demirci@studenti.unimi.it (T.N.D.); stefano.piazza@unimi.it (S.P.); enrico.sangiovanni@unimi.it (E.S.); giulia.martinelli@unimi.it (G.M.); 3Department of Science and Technology, University of Lille, Rue de Lille, 59160 Lille, France; eugenie.carpentiermaguire@gmail.com; 4Department of Lipid Science, CSIR-Central Food Technological Research Institute (CFTRI), Mysore 570 020, India; shilpa25188@gmail.com (S.T.S.); umamanjappara@cftri.res.in (U.V.M.); 5Institute of Medical Research and Medicinal Plants Studies (IMPM), Yaoundé 4123, Cameroon; armelle_d2002@yahoo.fr (A.D.T.); agogae@yahoo.fr (G.A.A.); 6Department of Pharmacy, University of Naples Federico II, 80131 Naples, Italy; maria.daglia@unina.it; 7International Research Center for Food Nutrition and Safety, Jiangsu University, Zhenjiang 212013, China; 8IRCCS MultiMedica, Sesto San Giovanni, 20099 Milan, Italy

**Keywords:** Cameroonian spice extracts, SW 872 adipocytes, triglyceride accumulation, glucose uptake, oxidative stress, pro-inflammatory cytokines

## Abstract

The molecular pathophysiology of cardiometabolic diseases is known to be influenced by dysfunctional ectopic adipose tissue. In addition to lifestyle improvements, these conditions may be managed by novel nutraceutical products. This study evaluatedthe effects of 11 Cameroonian medicinal spice extracts on triglyceride accumulation, glucose uptake, reactive oxygen species (ROS) production and interleukin secretion in SW 872 human adipocytes after differentiation with 100 µM oleic acid. Triglyceride content was significantly reduced by all spice extracts. Glucose uptake was significantly increased by *Tetrapleura tetraptera*, *Aframomum melegueta* and *Zanthoxylum leprieurii*. Moreover, *Xylopia parviflora*, *Echinops giganteus* and *Dichrostachys glomerata* significantly reduced the production of ROS. Concerning pro-inflammatory cytokine secretion, we observed that *Tetrapleura tetraptera*, *Echinops giganteus*, *Dichrostachys glomerata* and *Aframomum melegueta* reduced IL-6 secretion. In addition, *Xylopia parviflora*, *Monodora myristica*, *Zanthoxylum leprieurii*, and *Xylopia aethiopica* reduced IL-8 secretion, while *Dichrostachys glomerata* and *Aframomum citratum* increased it. These findings highlight some interesting properties of these Cameroonian spice extracts in the modulation of cellular parameters relevant to cardiometabolic diseases, which may be further exploited, aiming to develop novel treatment options for these conditions based on nutraceutical products.

## 1. Introduction

The use of well-characterized nutraceutical products, in addition to lifestyle changes, is an interesting option for prevention and management of chronic cardiovascular and metabolic diseases, including obesity, type 2 diabetes mellitus (T2DM), atherosclerosis and the metabolic syndrome [1], especially in milder pathological conditions [2,3]. In this context, dysfunction of the adipose tissue and derangement of adipocyte biology play a pivotal role involving several pathophysiological mechanisms including insulin resistance, oxidative stress, and low-grade chronic inflammation. Interestingly, many plant extracts used in traditional medicine were found to mitigate obesity and related dysmetabolic conditions in in-vitro experimental models as well as in clinical studies [4]. Our research group, along with other researchers, has previously investigated the composition and the molecular effects of several spice extracts from Cameroonian plants commonly used in traditional medicine. In particular, we explored the ability of these extracts to modulate the activity of some enzymes relevant to the control of cardiometabolic functions [5], the antioxidant and anti-inflammatory activities by gastric epithelial cells [6] and glucose uptake and antioxidant activity in human HepG2 cells [7].

Based on this knowledge, in this study we extended the evaluation of some potentially useful effects of these Cameroonian spice extracts in the context of adipocyte biology and pathophysiology. To address this issue, we took advantage of human SW 872 human liposarcoma cells differentiated to adipocytes by oleic acid (OA) as the model system, which previously proved suitable for testing the in-vitro metabolic activities of plant-derived extracts [8]. More specifically, the present study aimed at assessing the effects of these spice extracts on triglyceride (TG) accumulation, glucose uptake modulation, antioxidant activity and regulation of pro-inflammatory cytokine release.

## 2. Materials and Methods

### 2.1. Chemicals

Bovine insulin (Cat. No. I6634), oleic acid (OA) (Cat. No. O1257), (±)-6-Hydroxy 2,5,7,8 tetramethylchromane-2-carboxylic acid (Trolox) (Cat. No. 93510), dimethylsulfoxide (DMSO) (Cat. No. D8418), metformin hydrochloride (Cat. No. M0605000), epigallocatechin gallate (EGCG) (Cat. No. E4143), 2-deoxy-2-[(7-nitro-2,1,3-benzoxadiazol-4-yl) amino]-D-glucose (2-NBDG) (Cat. No. N13195) and resveratrol (Cat. No. R5010) were obtained from Merck Life Science (Readington, NJ, USA).

5-(and-6)-chloromethyl-2’,7’-dichlorodihydrofluorescein diacetate (CM-H2DCFDA) (Cat. No. C6827) was obtained from Thermo Fisher Scientific (Waltham, MA, USA). Fetal Bovine Serum (FBS) (Cat. No. ECS0180L) (Euroclone, Milan, Italy).

### 2.2. Preparation of Plant Extracts

Plant material comprised 11 spices (*Xylopia aethiopica* (Dunal) A.Rich., *Xylopia parviflora* (A. Rich) Benth, *Scorodophloeus zenkeri* Harms, *Echinops giganteus* A.Rich., *Monodora myristica* (Gaertn.) Dunal, *Tetrapleura tetraptera* (Schum. & Thonn.) Taub., *Dichrostachys glomerata* (Forssk.) Chiov., *Afrostyrax lepidophyllus* Mildbr., *Aframomum melegueta* K.Schum., *Aframomum citratum* (C.Pereira) K.Schum., and *Zanthoxylum leprieurii* Guill. & Perr.) harvested in different sites of the region of West Cameroon. Samples included fruits, seeds and roots, as identified in the National Herbarium of Cameroon (https://irad.cm/index.php/fr/, accessed on 14 October 2021) in Yaoundé (Cameroon) compared to the deposited specimens. The preparation of the hydroethanolic extracts was conducted as previously reported [5]. Briefly, air-dried and powdered samples (100 g) were extracted using a 70% hydroethanolic mixture at room temperature and in dark conditions. The mixture was filtered, concentrated under reduced pressure, and frozen to give crude extracts. They were then lyophilized and stored at −20 °C. Spice extract stock solutions (100 mg/mL) dissolved in DMSO were prepared, aliquoted, and kept at −80 °C. The stock solution of spice extracts in DMSO was diluted in culture medium to the appropriate concentrations for cell treatments. For more details, please check Table 1 in [7]. The same concentration of DMSO (in any case never exceeding 0.1%) was also added to all control cultures.

### 2.3. Cell Cultures and Differentation

The SW 872 human liposarcoma (ATCC^®^ HTB-92 ™) cell line was from the American Type Culture Collection (ATCC^®^, Manassas, VA, USA) and grown as recommended. Cells were cultured in DMEM-F12 culture medium with L-Glutamine and HEPES 25 mM supplemented with 10% fetal bovine serum (FBS), 1% penicillin (100 U/mL), and streptomycin (100 μg/mL). They were kept in culture in 100-mm diameter Petri dishes at 37 °C in a humidified atmosphere containing 5% CO_2_. Once they reached 80100% confluence, SW 872 cells were treated with 100 μM OA to initiate cellular differentiation into adipocytes [9,10]. 

### 2.4. Oil-Red-O Staining

To assess intracellular lipid accumulation (a marker of adipocyte differentiation) of SW 872 cells, we used the Oil-red-O (ORO) staining. SW 872 cells were seeded in 24-well plates, were allowed to adhere until 90–100%, confluence and then were treated with 100 μM OA for 7 days. After medium removal, cells were washed with phosphate-buffered saline (PBS) and fixed with 4% formaldehyde in PBS for 1 h (room temperature). ORO working solution (0.2% ORO in 40% isopropanol) was added to the culture dish and incubated at room temperature for 20 min. After a wash with sterile distilled water, cell images were collected with a light microscope (ZEISS, Milan, Italy).

### 2.5. MTS (3-(4,5-Dimethylthiazol-2-yl)-5-(3-carboxymethoxyphenyl)-2-(4-sulfophenyl)-2H-tetrazolium) Cell Viability Assay

Cell viability assay [11] was performed using the Cell Titer 96 aqueous non-radioactive cell proliferation assay (Promega, Madison, WI, USA) following the method previously described [12] with minor modifications. Briefly, SW 872 cells were seeded in a sterile, flat-bottom 96-well plate at a density of 2 × 10^5^ cells/well and incubated at 37 °C for 24 h in a humidified incubator containing 5% CO_2_. Extracts at different concentrations (1, 10, 25, 50, and 100 µg/mL) were prepared in fresh serum-free DMEM-F12 medium and 100 µL of each treatment was added to each well and then incubated for 24 and 48 h. Subsequently, 20 µL of the MTS reagent in combination with the electron coupling agent phenazine methosulfate was added to each well and allowed to react for 1 h at 37 °C. After 2 min of shaking at lowest intensity, the absorbance at 490 nm was measured (EnSpire PerkinElmer Multimode Plate Reader). In the same conditions, controls and blanks, consisting of cells with media containing DMSO (≤0.1%) and wells containing media without cells, respectively, were performed. The cell viability values were determined using the equation:(1)$$\%\;cell\;viability=\left(\frac{mean\;sample\;absorbance-mean\;blank\;absorbance}{mean\;control\;absorbance-mean\;blank\;absorbance}\right)\times 100$$

Three separate experiments run in triplicate replicates were conducted.

### 2.6. Morphological Analysis

Cells were cultured in sterile, flat-bottom 6-cm^2^ dishes for 24 h. Different concentrations of extracts (1, 10, 20 and 100 µg/mL) were prepared in serum-free media and 2 mL of each treatment was added to each dish and incubated for 24 h and 48 h. After treatment, cells were visualized with a light microscope (ZEISS, Milan, Italy) using 10×, 20× and 32× magnifications.

### 2.7. Interleukin-6 and Interleukin-8 Measurement

Interleukin-6 (IL-6) and interleukin-8 (IL-8) medium content was quantified using human IL-6 and IL-8 ELISA kits, as previously described [13]. Briefly, Corning 96-well EIA/RIA plates from Merck Life Science (Milan, Italy) were coated with the antibody provided in the ELISA Kit (Peprotech, London, UK) overnight at 4 °C. After blocking the reaction, 200 μL of samples in duplicate were transferred into wells and incubated at room temperature for 1 h. The amount of IL-6 and IL-8 in the samples was detected by the EnSpire Plate Reader (signal read: 450 nm, 0.1 s) using biotinylated and streptavidin–HRP conjugate antibodies, evaluating the 3,3′,5,5′ tetramethylbenzidine (TMB) substrate reaction. Quantification of IL-6 and IL-8 was done using an optimized standard curve provided with the ELISA Kit (8.0–1000.0 pg/mL). Resveratrol (10 μM) and EGCG (40 μM) were used as positive control for IL-6 and IL-8, respectively.

### 2.8. Triglyceride Content Measurement

To determine the accumulation of TG in the cells we used a sensitive assay using a TG quantification reagent (Vaktro Scientific, Patras, Greece). In this assay, TG are converted to free fatty acids and glycerol. Then, glycerol is oxidized to generate a colorimetric (570 nm)/fluorometric (λex = 535 nm/λem = 587 nm) product. The kit can detect 2 pmole–10 nmole (2–10,000 µM range) of TG. All samples and standards were measured in duplicate. The data obtained were interpolated in an appropriate triglycerides standard curve after background correction.

### 2.9. Measurement of ROS Production

The ntracellular ROS level in SW 872 cells were determined by a fluorometric assay, applying the oxidant-sensitive fluorescence probe CM-H_2_DCFDA according to the method described by Piazza et al. [14]. Cells were seeded in 96-well black plates at a density of 2 × 10^5^ cells/well, and after reaching 90% confluence, differentiated as previously described for 7 days with 100 µM oleic acid. Further, they were incubated with plant extracts at different concentrations ranging from 0 to 20 µg/mL for 24 h. Cells were then treated with 20 µM CM-H_2_DCFDA and then incubated for 1 h with 500 µM hydrogen peroxide to induce ROS production. The resulting fluorescence intensities were quantified at an excitation wavelength of 485 nm and an emission wavelength of 535 nm using the EnSpire Plate Reader (Perkin Elmer, Milan, Italy).

### 2.10. Glucose Uptake FACS Analysis

Differentiated and non-differentiated SW 872 cells were cultured in 12-well plates in DMEM F-12 + 10% FBS. After reaching about 80% confluence, cells were incubated with spice extracts (10–20 µg/mL) or insulin at different timing and concentrations. After incubation, cells were washed with PBS and treated with 20 µM 2-NBDG for 30 min in MEM Eagle w/Earle’s BSS without glucose. Finally, the 2-NBDG uptake was measured detecting the emitted fluorescence with FACS (BD LSRFortessa™ Flow Cytometer, Milan, Italy).

### 2.11. Western Blotting Analysis

Total protein extracts from SW 872 cells were obtained by lysing cells in 150 μL RIPA buffer containing a mix of protease and phosphatase inhibitors (Roche Diagnostics, Monza, Italy). Forty μg of proteins and a molecular mass marker (Novex^®^ Sharp Protein Standard, InvitrogenTM; Life Technologies, Monza, Italy) were separated on 10% sodium dodecylsulfate-polyacrylamide gel (SDS-PAGE) under denaturing and reducing conditions. They were then transferred to a nitrocellulose membrane by using the iBlotTM Gel Transfer Device (InvitrogenTM; Life Technologies). The membranes were washed with Tris-Buffered Saline-Tween 20 (TBS-T) and non-specific binding sites were blocked in TBS-T containing 5% BSA (Merck Life Science, Milan, Italy) or non-fat dried milk at room temperature for 60 min. The blots were incubated overnight at 4 °C with anti-pAKT(Ser473), (1:150; Millipore, Merck Life Science), anti-AKT (1:1000; Cell Signaling Technology, Milan, Italy), and anti β -Actin (Millipore, diluted 1:5000) (5% BSA or non-fat dried milk). Membranes were washed with TBS-T and then exposed for 30 min at room temperature to a diluted solution (5% non-fat dried milk) of the secondary antibodies. Immunoreactive bands were detected by exposure of the membranes to ClarityTM Western ECL chemiluminescent substrates (Bio-Rad Laboratories, Segrate, Italy) for 5 min and images were acquired with a ChemiDocTM XRS System (Bio-Rad Laboratories, Milan, Italy). Densitometric readings were assessed using the ImageLab software(6.1 version, Bio-Rad Laboratories, Milan, Italy).

### 2.12. Statistical Analysis

Results from at least three independent experiments carried out in triplicate were expressed as mean ±SD, values or as a mean percentage (%) compared to a control. Graphs and data were analyzed using GraphPad Prism version 8. The statistical analysis was made with one-way ANOVA followed by Tukey test, considering statistically significant *p* < 0.05.

## 3. Results

### 3.1. Spice Extracts up to 25 μg/mL Do Not Affect SW 872 Cell Viability and Morphology

Preliminary experiments were conducted to assess the effect of 11 Cameroonian spice extracts on the viability of differentiated SW 872 cells by MTS assay and morphological analysis. The extraction protocol, molecular composition, and potential cytotoxic effects in other cell lines of the spice extracts used in this study have been previously reported [5,6,7]. The MTS assay was conducted after treatment with different concentrations (1–100 μg/mL) of extracts for 24 h and 48 h. Viability threshold was set at 80% [15]. All spice extracts incubated for 24 h or 48 h proved non toxic up to the concentration of 25 μg/mL (Appendix A). More specifically, after 24 h of incubation, 6/11 extracts proved toxic at 50 μg/mL and after 48 h 2/11 plants were toxic already at 25 μg/mL. Toxicity at 100 μg/mL was observed for 10/11 extracts in SW 872 cells, independently of the incubation time (Appendix A). We also investigated whether spice extracts affected cell morphology in addition to viability. The morphology of differentiated SW 872 cells was not affected by treatment for 24 h and 48 h with 1–25 μg/mL spice extracts. At 100 μg/mL, the maximum concentration tested, all but one (*A. lepidophyllus*) spice extracts produced morphological changes, including cell death, rounding, and shrinking (representative images obtained with *X. aethiopica* and *A. lepidophyllus* extracts are shown in Appendix A). Based upon these results obtained by MTS assay and morphological analysis, all treatments with spice extracts were conducted for 24 h or 48 h at the concentration of 20 μg/mL or lower.

### 3.2. All Spice Extracts Reduce Triglyceride Accumulation in Differentiated Adipocytes

SW 872 cells were differentiated to adipocytes with 100 µM OA for seven days, according to a previously published protocol [8,9]. This treatment led to a marked intracellular lipids accumulation, as observed by ORO staining analysis (Figure 1) and quantification of TG content (Table 1), indicating the acquisition of a mature adipocyte phenotype. Of interest, differentiation with OA resulted in a doubling of TG accumulation (*p* < 0.05), compared to untreated cells (Table 1). We then assessed whether treatment with spice extracts was able to modulate TG accumulation in differentiated SW 872 cells, which were treated for 24 h and 48 h with spice extracts (all at 10 µg/mL). TG content did not change after a 24 h exposure to extracts. However, after 48 h, all extracts were able to significantly (*p* < 0.05) decrease TG accumulation (−11.3%/−18.5%). Resveratrol (10 µM), taken as active control, significantly (*p* < 0.05) reduced TG accumulation by 30.4% (Table 1). We then assessed whether treatment with spice extracts was able to modulate TG accumulation in differentiated SW 872 cells, which were treated for 24 h and 48 h with spice extracts (all at 10 µg/mL). TG content did not change after a 24 h exposure to extracts. However, after 48 h, all extracts were able to significantly (*p* < 0.05) decrease TG accumulation (−11.3%/−18.5%). Resveratrol (10 µM), taken as active control, significantly (*p* < 0.05) reduced TG accumulation by 30.4% (Table 1).

### 3.3. Tetrapleura tetraptera, Aframomum melegueta and Zanthoxylum leprieurii Increase Glucose Uptake in Differentiated Adipocytes

Glucose uptake is a pivotal event of adipocyte biology [16] and was assessed here in both undifferentiated and differentiated SW 872 cells by FACS analysis, measuring the uptake of 2-NBDG. Basal glucose uptake was significantly reduced (−54.8% vs controls; *p* < 0.001) upon OA-differentiation compared to undifferentiated controls (Figure 2A). Insulin-activated glucose uptake was assessed according to time-course (15–60 min) and dose-response (10 nM–1 µM) parameters (Appendix A). A significant increase of glucose uptake was however observed in differentiated cells after 60 min incubation with 10 nM (+16.8%; *p* < 0.05) and 100 nM (+18.8%; *p* < 0.01) insulin (Appendix A). The uptake of 2-NBDG was significantly (*p* < 0.001) increased by *T. tetraptera* (+40.8%), *A. melegueta* (+41.7%) and *Z. leprieurii* (+56.6%), all at the concentration of 10 µg/mL (Figure 2B,C). This increment was even greater than that elicited by 100 nM insulin (+18.8%; *p* < 0.01). To test whether such increase was dose dependent, we selected the three most effective extracts and added *S. zenkeri* since in preliminary experiments this spice extract was effective at 20 µg/mL (Figure 2D).

We then explored the activation of the Akt signalling pathway, known to be associated with insulin receptors, under non-differentiated and OA-differentiated conditions and observed that 100 nM insulin treatment elicited a rapid and prolonged phosphorylation of Akt in OA-differentiated cells but not in non-differentiated cells (Figure 3A). This insulin-driven activation of Akt was not affected by a 24-h pretreatment of SW 872 cells with the three extracts that were found more effective in promoting glucose uptake (*A. melegueta*, *Z. leprieurii,* and *T. tetraptera*) (Figure 3B). 

### 3.4. Spice Extracts Modulate IL-6 and IL-8 Release from Differentiated Adipocytes

The release of pro-inflammatory cytokines, like IL-6 and IL-8, is a central pathophysiological feature of disfunctional adipocytes [17]. OA-differentiation of SW 872 cells resulted in a significant (*p* < 0.001) increase of IL-6 release (Figure 4A), which was significantly reduced by exposure (24 h, 10 µg/mL) to *A. melegueta* (−43.1%; *p* < 0.001), *D. glomerata* (−39.9%; *p* < 0.001), *T. tetraptera* (−29.7%; *p* < 0.05) and *E. giganteus* (−29%; *p* < 0.05) (Figure 4B). Resveratrol, selected as positive control, resulted in a 67.5% (*p* < 0.001) reduction (Figure 4B). Conversely, IL-8 release was significantly reduced (*p* < 0.01) upon OA-differentiation of SW 872 cells (Figure 4C). IL-8 secretion was further decreased by exposure to *X. parviflora,* (−36.8%; *p* < 0.001), *X. aethiopica* (−21.1%; *p* < 0.05)*, M. myristica* (−24.3%; *p* < 0.01) and *Z. leprieurii* (−32.7%; *p* < 0.01) (Figure 4D). On the contrary, treatment with *D. glomerata* and *A. citratum* resulted in a marked increase of IL-8 release (+58.6% and +78.7%, respectively; *p* < 0.001).

### 3.5. Spice Extracts Affect Reactive Oxygen Species Production in Differentiated Adipocytes

The protective effect of antioxidants on cells is related to their ability to reduce the level of intracellular ROS generation. To assess whether spice extracts could reduce intracellular oxidative stress, intracellular ROS generation was quantified in differentiated SW 872 cells using the CM-H_2_DCFDA assay. Cells were differentiated with OA for seven days and treated or not for another 24 h with spice extracts before being exposed for 1 h to 500 µM H_2_O_2._ Treatment with 500 µM H_2_O_2_ alone produced in a 2.5-fold increase in intracellular ROS levels (Table 2).

When cells were pre-treated for 24 h with spice extracts (all at 10 µg/mL), ROS production induced by 1 h treatment with H_2_O_2_ was reduced to a different extent by all spices except *A. citratum, Z. leprieurii,* and *S. zenkeri,* which were ineffective. Moreover, *X. aethiopica* and *A. melegueta* extracts significantly increased ROS production (Table 2). In addition, the most effective extracts (*X. parviflora, E. giganteus, and D. glomerata*) were found to exert their ROS scavenging activity in a concentration-dependent fashion (Figure 5).

## 4. Discussion

Nutraceutical approaches to prevent and treat cardiometabolic conditions are nowadays a relevant and promising research field [18], in some instances taking advantage of knowledge derived from traditional medicine. In the present study, we evaluated the effects of a set of plant extracts used in Cameroon as nutritional spices and medicinal agents [19,20] on cellular events related to the molecular pathophysiology of cardiometabolic diseases. In consideration that ectopic adipose tissue, especially when dysfunctional, is a well-known important player in cardiovascular and metabolic diseases [21], in the present study, we selected human SW 872 differentiated adipocytes as the model system [22,23]. SW 872 cells show, under basal conditions, an immature phenotype and constitutively express several genes involved in fatty acid metabolism, such as lipoprotein lipase (LPL), cholesterol ester transfer protein (CETP), cluster of differentiation 36 (CD36), peroxisome proliferator-activated receptor (PPAR)α, PPARγ and LDL receptor-related protein (LRP) [22]. For the purpose of this study, SW 872 were differentiated to mature adipocytes using OA, according to established protocols [9]. We observed an overall inhibitory action across all plant extracts regarding TG content of SW 872 cells since the massive TG accumulation induced by OA treatment was reduced by −11/−18% by all extracts after 48 h. Consistent with our data, it has been reported that lipid accumulation was significantly decreased in 3T3-L1 adipocytes by resveratrol [24], which was selected here as a positive control. All the tested spice extracts are rich in a variety of primary metabolites (glycerol, fructofuranose, glucopyranose, etc) and secondary metabolites (chlorogenic acid, catechin, pimaric acid, etc.) [5] that orchestrate the observed molecular responses.

The occurrence of some plant specificity in this effect is suggested by our previous observation that *Adansonia digitata* L. extracts did not affect TG accumulation in differentiated SW 872 adipocytes [8]. The development of obesity is accompanied by adipocyte hypertrophy and hyperplasia [25] leading to excess of lipid accumulation [26], and the observed effects may at least in part explain the actions of some of the tested plants in animal models. For example, extracts from *A. melegueta* seeds were found to reduce adipose tissue in obese mice [19], and high-carbohydrate and high-fat diet-induced obesity and diabetes in rats were attenuated by *T. tetraptera* extract [27]. Dysfunctional adipocytes, as observed in T2DM and obesity, also develop reduced glucose uptake [28,29,30]. In this study, *T. tetraptera, A. melegueta* and *Z. leprieurii* increased glucose uptake in a dose-dependent manner. Interestingly, in another in-vitro model, the HepG2 cells, *T. tetraptera* and *A. melegueta,* but not *Z. leprieurii* increased glucose uptake [7], underlining the relationship between the complex molecular composition of the spice extracts, the observed effect and the specificity linked to the tested cell model. In differentiated SW 872 cells, the tested spice extracts do not appear to modulate the Akt upstream pathway whether or not in the presence of insulin, suggesting that they may increase glucose uptake by promoting other molecular events, like glucose transporter translocation to the membrane. Moreover, insulin resistance, glucose uptake impairment and metabolic disfunction appear to be sustained by the secretion from dysfunctional adipocytes of several pro-inflammatory cytokines [31] as well as by the resultant ROS production [1,32]. Based on these considerations, we evaluated the secretory profile of IL-6 and IL-8, two NF-kB-dependent molecular mediators released in the context of chronic inflammation [7]. Among the 11 tested extracts, four (*A. melegueta, D. glomerata, T. tetraptera and E. giganteus*) were effective in significantly reducing (−29/−43%) IL-6 release. *T. tetraptera,* and *D. glomerata* were also found to similarly reduce IL-6 release by GES-1 gastric epithelial cells [6]. In addition, four spice extracts (*X. parviflora, X. aethiopica, M. myristica,* and *Z. leprieurii*) were able to reduce (−21/−36%) IL-8 release, whereas this was markedly increased by *D. glomerata* and *A. citratum*. Notably, *D. glomerata* extract showed a dichotomic effect, reducing IL-6 and increasing IL-8 secretion. It is important to highlight that the release of each cytokine was negatively modulated by fully different sets of plants. Although it is challenging to exactly establish the role of each extract and their metabolites in IL-6 and IL-8 modulation, it is well known that shogaol, gingerol, chlorogenic acid, pimaric acid, and other molecules identified in our extracts [5] are well known inhibitors of NF-κB [14,33,34,35,36].

Furthermore, we assessed the effects of treatment with these Cameroonian spice extracts on H_2_O_2_-induced ROS production in differentiated SW 872 cells. All extracts, except *A. citratum, Z. leprieurii* and *S. zenkeri* were able to reduce ROS production. *X. parviflora, E. giganteus* and *D. glomerata* were the most effective and showed a dose-related activity. These antioxidant effects could be explained by the high content in phenolic compounds and the potent in-vitro antioxidant capacity previously reported by our group [7]. ROS production was however increased by *X. aethiopica* and *A. melegueta* extracts.

Taken together, the findings of the present study, summarized in Table 3, suggest that each of these spice extracts display a rather peculiar profile of activity on differentiated SW 872 adipocytes. This deserves further exploitation, in order to highlight in a more robust way the peculiar utilization of each spice extract to target specific functions (i.e., antioxidant vs. anti-inflammatory or promoting glucose uptake). Nutraceutical products with these activities may find their application in subjects with obesity, metabolic syndrome, T2DM and the related atherosclerotic cardiovascular disease risk, either alone, especially in milder conditions [37], or in combination with selected drugs in order to avoid drug dosage increase, prevent some adverse effects and increase the overall efficacy [4,38]. Interestingly, some pathophysiological mechanisms underlying these conditions, like oxidative stress and chronic low-grade inflammation, do not seem properly managed by the current pharmacology, and thus, well-characterized plant extracts could allow to specifically target them.

This study has some limitations. OA-differentiated SW 872 cells resemble in some respects human adipocytes although this differentiation protocol is associated with peculiar molecular effects related to PPARγ activation, which therefore need to be taken into consideration in the interpretation of the obtained results. It is also important to underline that the extraction protocol utilized in this study [5] may differ from that used by others, making in some cases the comparison with other studies on the same plants difficult. This work, together with the others published in the last years, contributes to define and enrich the study of the biological activities of a set of Cameroonian spices, particularly in the context of dysfunctional adipocyte cell biology.

## 5. Conclusions

The results of this study show that the tested Cameroonian spice extracts display interesting activities on several molecular features of differentiated SW 872 human adipocytes. All extracts were able to reduce TG accumulation, while the ability to promote glucose uptake, reduce pro-inflammatory cytokines release, and counteract ROS production was limited to panels of three to six plant extracts (Table 3). Such variety of effects may be the basis for the development of nutraceutical products with very specific effects or combining more than one extract to achieve complementary effects. Moreover, *T. tetraptera* stands out as the most versatile plant since it was found to positively modulate most parameters. Since the 11 spice extracts showed rather variable viability profiles, with some extracts being toxic at concentrations just higher than the most effective doses (10–20 µg/mL), a word of caution should be spent regarding the potential toxicity of some plants when used at higher concentrations and/or prolonged treatments.

The findings of the present study, conducted in a human adipocyte in-vitro model, highlight some potential health properties of these Cameroonian spices and suggest the opportunity of further studies in the context of experimental and human cardiometabolic diseases.

## Figures and Tables

**Figure 1 nutrients-13-04271-f001:**
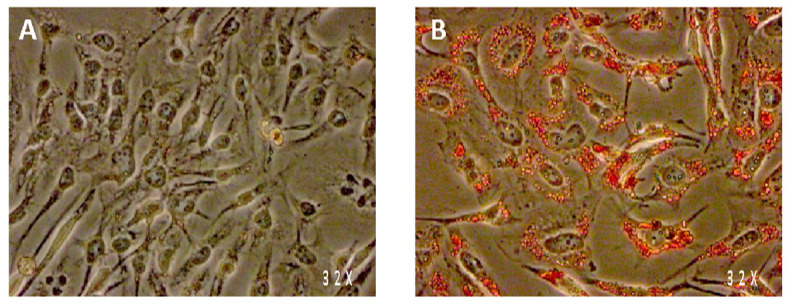
Intracellular lipid content (Oil-red-O staining) of SW 872 cells after 7 days of incubation without any treatment (**A**) or with 100 µM oleic acid (**B**).

**Figure 2 nutrients-13-04271-f002:**
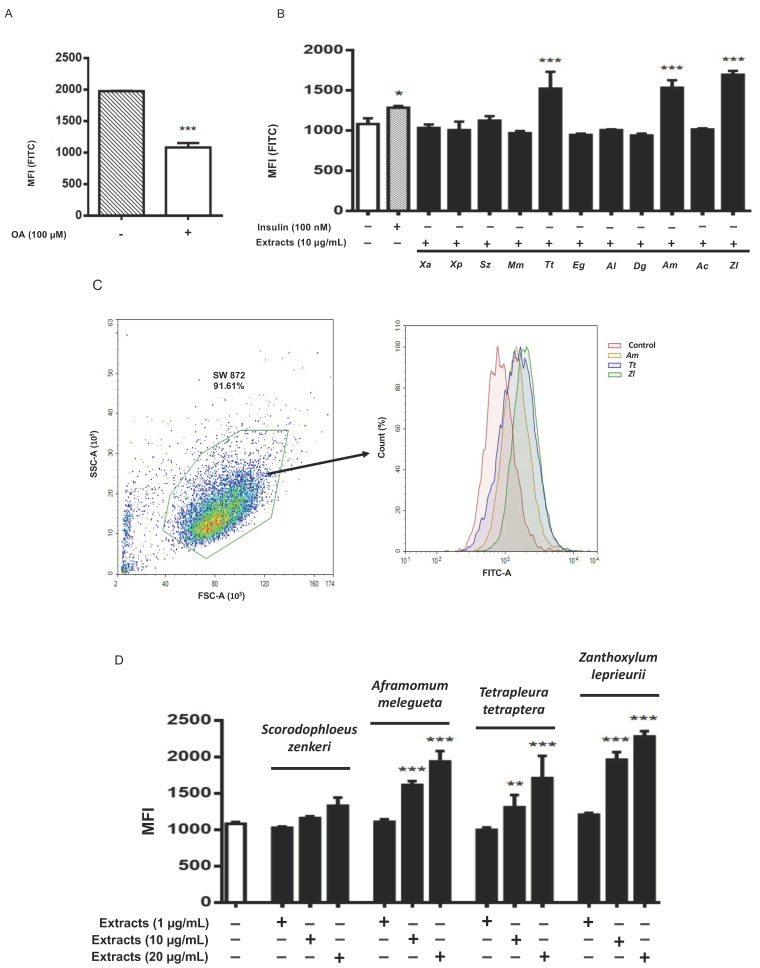
Glucose uptake in SW 872 cells treated with spice extracts. (**A**) Basal glucose uptake in non-differentiated and differentiated cells (100 µM OA); *** *p* < 0.001 (unpaired *t*-test). (**B**) Differentiated cells were incubated for 1 h with 100 nM insulin or treated for 24 h with 10 µg/mL of each spice extract. *Xa*, *Xylopia aethiopica*; *Xp*, *Xylopia parviflora*; *Sz*, *Scorodophloeus zenkeri*; *Mm*, *Monodora myristica*; *Tt*, *Tetrapleura tetraptera*; *Eg*, *Echinops giganteus*; *Al*, *Afrostyrax lepidophyllus*; *Dg*, *Dichrostachys glomerata*; *Am*, *Aframomum melegueta*; *Ac*, *Aframomum citratum*; *Zl*, *Zanthoxylum leprieurii*. * *p* < 0.05, *** *p* < 0.001. (**C**) Mean fluorescence intensity (MFI) of FACS analysis obtained in differentiated SW 872 cells by treatment with *Aframomum melegueta* (*Am*), *Zanthoxylum leprieurii* (*Zl*) and *Tetrapleura tetraptera* (*Tt*). The control (red area) is represented by differentiated SW 872 cells MFI. (**D**) Differentiated cells were treated for 24 h with the indicated extracts at 1–10–20 µg/mL. Glucose uptake was assessed in SW 872 cells by FACS analysis. One experiment (*n* = 3) is shown as representative of 3 separated experiments, each in triplicate. Results are expressed as mean ± SD. ** *p* < 0.01, *** *p* < 0.001 (one-way ANOVA multiple comparison).

**Figure 3 nutrients-13-04271-f003:**
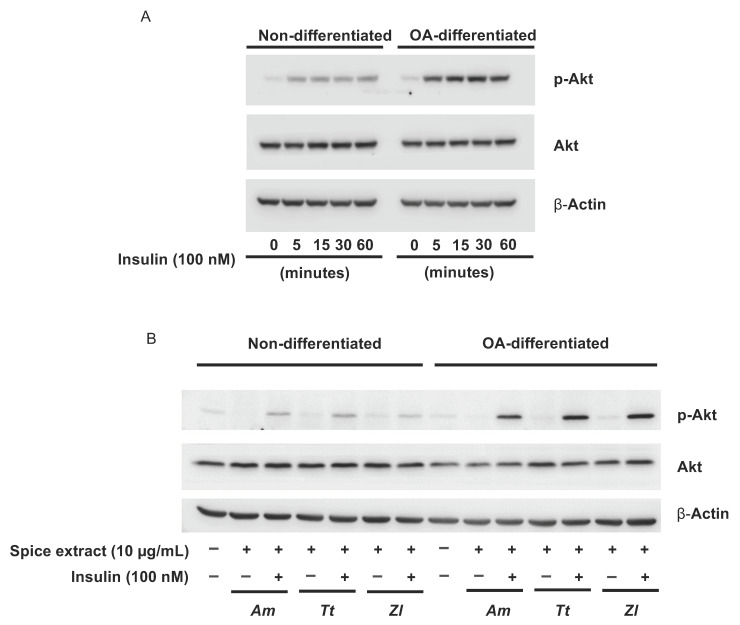
Akt phosphorylation dynamics in differentiated and non-differentiated SW 872 cells. (**A**) Time course of Akt phosphorylation upon treatment with 100 nM insulin. (**B**) Combined effect of 24-h exposure to 10 µg/mL spice extracts (*A. melegueta* (*Am*), *Z. leprieurii* (*Zl*) and *T. tetraptera* (*Tt*)) and 1-h treatment with 100 nM insulin on Akt phosphorylation.

**Figure 4 nutrients-13-04271-f004:**
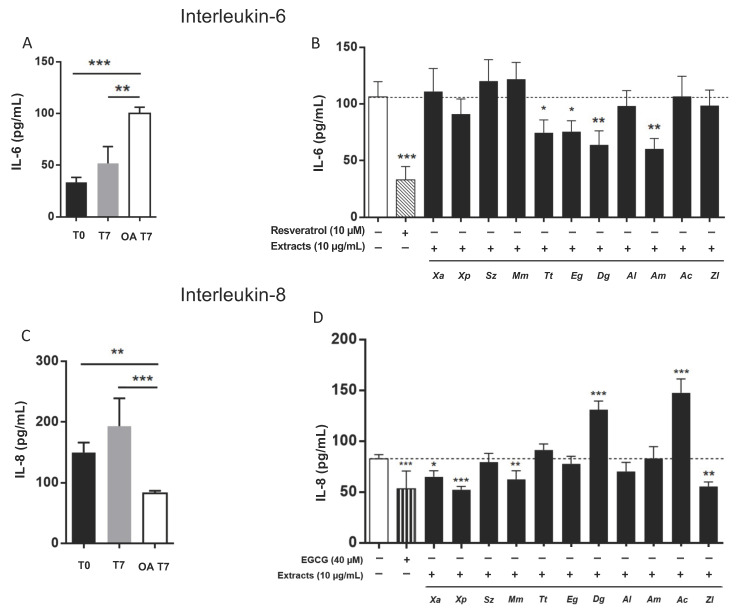
Effects of spice extracts on interleukin-6 (IL-6) and interleukin-8 (IL-8) release by differentiated SW 872 cells. (**A**) Basal IL-6 content was determined at T0 and T7 (1 and 7 days after seeding, respectively) and at OA T7 (after 7-day differentiation with 100 µM oleic acid (OA)). (**B**) OA-differentiated cells were treated with 10 µg/mL spice extracts or the positive controls (10 µM resveratrol or 40 µM epigallocatechin gallate (EGCG)). (**C**) Basal IL-8 content was determined at T0 and T7 (1 and 7 days after seeding, respectively) and at OA T7 (after 7-day differentiation with 100 µM OA). (**D**) OA-differentiated cells were treated with 10 µg/mL spice extracts or the positive controls (10 µM resveratrol or 40 µM EGCG). IL-6 and IL-8 content in the culture medium was determined after 24 h incubation. One experiment (*n* = 3) is shown as representative of 3 separate experiments, each in triplicate. Results are shown as mean ± SD. Data are expressed as pg/mL. * *p* < 0.05, ** *p* < 0.01, *** *p* < 0.001 (one-way ANOVA multiple comparison). *Xa*: *Xylopia aethiopica*; *Xp*: *Xylopia parviflora*; *Sz*: *Scorodophloeus zenkeri*; *Mm*: *Monodora myristica*; *Tt*: *Tetrapleura tetraptera*; *Eg*: *Echinops giganteus*; *Dg*: *Dichrostachys glomerata*; *Al*: *Afrostyrax lepidophyllus*; *Am*: *Aframomum melegueta*; *Ac*: *Aframomum citratum*; *Zl*: *Zanthoxylum leprieurii*.

**Figure 5 nutrients-13-04271-f005:**
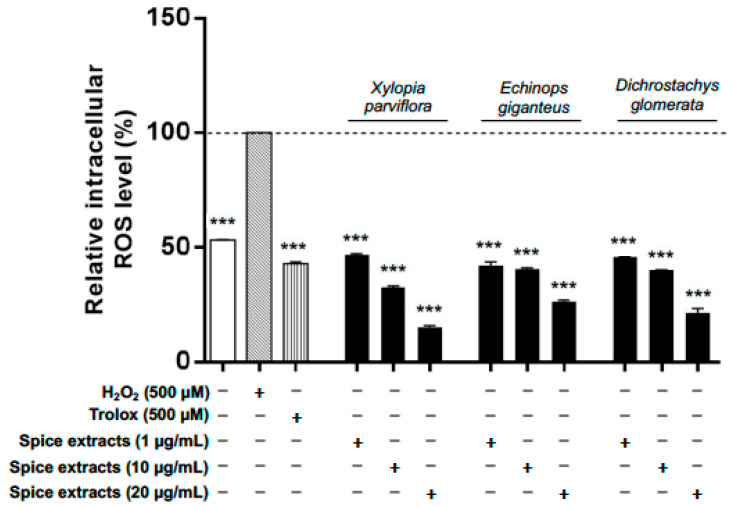
Concentration-dependent modulation of intracellular ROS release in SW 872 cells. The 3 most potent spice extracts were selected to evaluate the dose response. Concentration-dependent reduction of intracellular ROS production in SW 872 cells by selected spice extracts (*Xylopia parviflora, Echinops giganteus,* and *Dichrostachys glomerata*). Data are expressed as % of control taken as 100; mean ±SD, *n* = 3. One experiment (*n* = 3) is shown as representative of 3 separated experiments, each in triplicate. Results are expressed as mean ±SD. *** *p* < 0.001 (one-way ANOVA multiple comparison).

**Table 1 nutrients-13-04271-t001:** Triglyceride accumulation in differentiated SW 872 adipocytes: effect of spice extracts.

	Oleic Acid (100 µM)	Triglyceride (% of Oleic Acid-Treated Cells)	
		+24 h	+48 h
undifferentiated control	−	49.6 ± 4.1 ***	64 ± 0.7 ***
differentiated control	+	100	100
resveratrol (10 µM)	+	94.9 ± 5.9	69.6 ± 3.0 ***
*Xylopia aethiopica*	+	96.5 ± 7.6	85.5 ± 2.3 *
*Xylopia parviflora*	+	88.2 ± 1.0	86.2 ± 1.6 *
*Scorodophloeus zenkeri*	+	97.9 ± 9.6	81.5 ± 4.9 *
*Monodora myristica*	+	90.4 ± 9.0	84.7 ± 3.2 *
*Tetrapleura tetraptera*	+	90.9 ± 4.5	86.2 ± 2.2 *
*Echinops giganteus*	+	104.2 ± 5.9	88.7 ± 0.9 *
*Afrostyrax lepidophyllus*	+	102.5 ± 6.4	83.5 ± 10.1 *
*Dichrostachys glomerata*	+	102.6 ± 2.5	82.6 ± 6.8 *
*Aframomum melegueta*	+	104 ± 5.2	87.0 ± 4.7 *
*Aframomum citratum*	+	98.3 ± 4.7	84 ± 1.8 *
*Zanthoxylum leprieurii*	+	98.5 ± 8.8	86.6 ± 5.9 *

All spice extracts were used at 10 µg/mL. Data are expressed as % of oleic acid-treated cells taken as 100; mean ± SD, *N* = 3, (*p* ˂ 0.05) (one-way ANOVA multiple comparison); * *p* < 0.05, *** *p* < 0.001 vs. the differentiated control group.

**Table 2 nutrients-13-04271-t002:** Intracellular ROS production in differentiated SW 872 adipocytes: effect of spice extracts.

	H_2_O_2_ (500 µM)	Relative Intracellular ROS Level (%)
Control	−	35.9 ± 0.2 ***
H_2_O_2_ (500 µM)	+	100
Trolox (500 µM)	+	61.8 ± 8.6 ***
*Xylopia aethiopica*	+	255.7 ± 9.3 ***
*Xylopia parviflora*	+	49.6 ± 3.53 **
*Scorodophloeus zenkeri*	+	94.7 ± 1.6
*Monodora myristica*	+	60.0 ± 8.9 ***
*Tetrapleura tetraptera*	+	72.7 ± 1.5 **
*Echinops giganteus*	+	56.4 ± 1.2 ***
*Afrostyrax lepidophyllus*	+	75.4 ± 0.9 *
*Dichrostachys glomerata*	+	66.6 ± 0.6
*Aframomum melegueta*	+	122.5 ± 9.9 **
*Aframomum citratum*	+	112.8 ± 0.4
*Zanthoxylum leprieurii*	+	89.9 ± 3.3

All plant extracts were used at 10 µg/mL. Data are expressed as % of H_2_O_2_-treated cells taken as 100; mean ± SD, *n* = 3. One experiment (*n* = 3) is shown as representative of 3 separated experiments, each in triplicate. Results are expressed as mean ± SD. * *p* < 0.05, ** *p* < 0.01, *** *p* < 0.001 (one-way ANOVA multiple comparison).

**Table 3 nutrients-13-04271-t003:** Summary of the specific effects of the tested Cameroonian spice extracts on differentiated SW 872 adipocytes.

	Triglyceride Reduction	Glucose Uptake Stimulation	ROS Production	IL-6 Reduction	IL-8 Reduction
*Xylopia aethiopica*	−14.5%		+55.8%		−21.1%
*Xylopia parviflora*	−13.8%		−50.5%		−36.8%
*Scorodophloeus zenkeri*	−18.5%				
*Monodora myristica*	−15.3%		−40%		−24.3%
*Tetrapleura tetraptera*	−13.8%	+40.8%	−27.4%	−29.7%	
*Echinops giganteus*	−11.3%		−43.6%	−29%	
*Afrostyrax lepidophyllus*	−16.5%		−24.6%		
*Dichrostachys glomerata*	−17.4%			−40%	
*Aframomum melegueta*	−13%	+41.7%		−43.1%	
*Aframomum citratum*	−16%				−58.6%
*Zanthoxylum leprieurii*	−13.4%	+56.6%			−32.7%

## Data Availability

The data presented in this study are available on request from the corresponding authors.

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
