# Peer review of "Cameroonian Spice Extracts Modulate Molecular Mechanisms Relevant to Cardiometabolic Diseases in SW 872 Human Liposarcoma Cells"

_nutrients, 2021, doi:10.3390/nu13124271_

Round 1

Reviewer 1 Report

DMSO was used as a solvent by the researchers. In the Discussion, I am wondering about the need to add how the methods took into account that DMSO has anti-inflammatory properties. The literature includes several studies that associate DMSO intake with declines in inflammatory biomarker activity, including IL-6 and IL-8. DMSO has also been found to be an antagonist to lipid cells. Given the researchers measured the same biomarkers, was the use of DMSO as a solvent a confounding variable? If so, did statistical analysis adjust for this potential confounder?

Author Response

Dear Reviewer, thanks for your comments, which have been addressed as reported in the attached file.

Paolo Magni

Reviewer 2 Report

The manuscript entitled ‘Cameroonian spice extracts modulate molecular mechanisms relevant to cardiometabolic diseases in SW872 human liposarcoma cells’ by Nwakiban et al, is an investigation of 11 Cameroonian spice extracts on triglyceride accumulation, glucose uptake and ROS production. To this end, the authors utilized the liposarcoma cell line SW872 and treated with the extracts and studied cellular parameters.

Comments

  1. A major limitation of the study is that authors have just utilized one cell line in their study. It is understandable that for certain studies related to triglyceride accumulation, such a model is useful. However, the other parameters should be tested in at least another cell model. Authors have used a cancer cell model and tried to implicate the findings for non-cancer patients. Line 395, the discussion of the potentials of the findings are very vague. Authors suggest the use of well-characterized plant extracts, but there is no discussion of the potential active components in the extracts. The authors have previously published on the phenolic nature of the compounds, but should also discuss it with current results.
  2. Supplementary data are not provided. Without that data, it is difficult to judge the manuscript.
  3. Material and methods, please provide catalog numbers. And authors have mentioned in bracket city, country of the related company. What does it mean? Merck Life Science (Milan, Italy). Often the head office of the companies are in different countries.
  4. References, are not mentioned correctly everywhere. In some places, author et al and other only the number. Please follow a consistent method.
  5. The authors have referred to previous publication about how the extracts were prepared. They should also mention it here briefly.
  6. Table 1, related to this table, do the authors have pictures to show of the Oil Red staining?
  7. Table 1, authors here used Resveratrol at 10uM, but the associated reference has found significant results at higher conc of 25 to 50uM. These results are not very convincing. The associated reference shows a concentration where there is strong decrease in the lipid content, but this is not visible here with any of the extracts.
  8. The investigation of pAkt by the extracts looks like a negative result. The authors show that some extracts increase glucose uptake, implicate some insulin receptor, but show negative result on pAkt. The conclusion of this experiment is not clear.
  9. The motivation for studying IL6 and IL8 is not very clear. Authors say that their increased levels are features of dysfunctional adipocytes. This statement is unclear.

Author Response

(The authors gave the same response as above.)
